# Toughening and Hardening Limited Zone of High-Strength Steel through Geometrically Necessary Dislocation When Exposed to Electropulsing

**DOI:** 10.3390/ma15175847

**Published:** 2022-08-24

**Authors:** Yunfeng Xiong, Zongmin Li, Tao Liu

**Affiliations:** School of Marine Engineering, Jimei University, Xiamen 361016, China

**Keywords:** weld heat-affected zone, electropulsing treatment, local toughening and hardening, tailoring martensitic substructure

## Abstract

The enhancement of both low-temperature impact toughness and the hardness of a high strength steel heat-affected zone (HAZ) is investigated by using high-density electropulsing (EP). The athermal and thermal effects of EP on HAZ microstructure and resultant mechanical properties were examined based on physical metallurgy by electron backscattered diffraction and on tests of hardness and impact toughness at −60 °C, respectively. EP parameters were carefully determined to avoid electro-contraction and excessive pollution of the base metal by using numerical simulation. The EP results show that the mean impact toughness and hardness of HAZ are 2.1 times and 1.4 times improved, respectively. In addition to the contribution of microstructure evolution, geometrically necessary dislocation (GND) is also a contributor with an increase of 1.5 times, against the slight decrease in dislocation line density and dislocation density. The mechanisms behind this selective evolution of dislocation components were correlated with the localized thermal cycle EP, i.e., the competition among thermo- and electro-plasticity, and work-hardening due to local thermal expansion. The selective evolution enables the local thermal cycle EP tailor the martensitic substructure that is most favorable for toughness and less for hardness. This selective span was limited within 4 mm for a 5 mm thick sample. The local thermal cycle EP is confirmed to be capable of enhancing in both toughness and hardness within a millimeter-scale region.

## 1. Introduction

High-strength steel (HSS) is becoming popular for weight-saving construction, especially in oceanic engineering. Welding is an effective technique for connecting components, but the introduction of thermal gradients to joints leads to highly heterogeneous microstructures in heat-affected zones (HAZs). The coarse-grained heat-affected zone (CGHAZ) is a resultant sub-zone within the HAZ and is typically susceptible to being brittle and fracturing [1].

Post-weld heat treatment is a general alloying-free process used for enhancing the HAZ. Studies have focused on adjusting the heat-treatment parameters, such as the holding time at the intercritical temperature [2] and cooling rate [3]. General heat treatment readily generates a heat-affected region beyond that of the HAZ, thus contaminating the adjacent base metal.

Electropulsing (EP) is an alternative potential for less pollution due to its high density of energy. Previous studies have focused on modifying metal mechanical properties, i.e., softening and strengthening, termed thermal and athermal effects according to EP-introduced temperature elevation T. If the T is far below phase transformation, the EP effect is dominant with an athermal effect, i.e., electro-plasticity [4]. In this respect, considerable softening has been reported because of coarse microstructure [5] or release of residual stress [6]. When T is high enough to facilitate phase transformation, strengthening becomes dominant with the thermal effect, i.e., rapid Joule heating and cooling [7]. However, an approximate T meeting the demand of simultaneous toughening and hardening by using EP has still been seldom reported, in which the competition between the thermal aspect, i.e., strengthening, and the athermal aspect, i.e., softening, is not clear.

Another challenge in EP usage lies in the uniformity of temperature elevation through thickness arising from the electro–contraction effect when a thick workpiece is present. Previous studies have focused on special sample configurations: (i) thin samples, such as those with a thickness ranging from 1 mm [8,9] to 2.64 mm [5]; and (ii) samples with shrinkage cross-sections, such as the dog bone shape [8,9] or a wider-end shape [5]. Those sample configurations facilitate the EP design regardless of electro–contraction effect. With the disabling of the shape configuration of a thick HAZ, a proper strategy of EP is most concerned with meeting the demands of both the homogeneous microstructure through thickness and the need for less pollution of the adjacent microstructure.

In this study, the EPT strategy was carefully examined to balance overheating due to the consideration of homogeneous Joule heating through thickness and excessive pollution of the adjacent microstructure. The resultant competition between softening and strengthening was elaborated. The findings of this study provide fundamental insights into the local toughness and hardness of HSS by taking advantage of the coupled thermal and athermal effects of EP. Such research has not been conducted before, and therefore, could provide novel insights into engineering enhancement of both hardness and toughness by EP.

## 2. Materials and Experiment

Before impact testing and microstructure characterization, EPT was carried out on HAZ. The parameters of EPT were carefully checked to find out the problem of EPT for a thick sample.

### 2.1. Materials and Weld Procedure

FH690 ship steel with a microstructure of ferrite and pearlite was used in this study. Its composition was 0.08 C, 0.15 Si, 1.58 Mn, 0.8 Ni, 0.135 Nb + V + Ti, 0.002 S, 0.003 P, and balanced Fe (in wt.%). Its mechanical properties were measured to be a yield strength of 690 MPa, a toughness of 46 J and a hardness of 280 HV. Its starting temperature (Ac_1_) and completion temperature (Ac_3_) for the austenitization were estimated to be 699 °C and 847 °C [10], respectively.

Plates of 15 mm thickness were joined by hybrid gas(20% Ar and 80% CO_2_)-shielded arc welding with a Megafil 690R E110-111T1 (AWS code, φ1.2 mm) weld filler. A square-groove as wide as 8 mm was configured to generate two comparable microstructures of HAZs along the plate normal direction (ND) under the same welding process, one for the impact test and the other for microstructure observation, as shown in Figure 1.

To promote the formation of the coarse-grained HAZ microstructure, three-pass beads were deposited, allowing for sufficient austenite growth above Ac_3_ with parameters as follows: current: 110–120 A; arc voltage: 20–22 V; and gas flow rate: 22 L/min. The temperature between each weld pass was maintained between 100 and 150 °C, as monitored by a non-contact infrared thermometer. Figure 2a,b shows the optical microstructures of HAZ and weld bead, consisting of a mixture of bainite–ferrite (BF), pearlite (P) and martensite (M), respectively. As a typical characteristic of HAZ, prior austenite grain boundaries (PAGBs) can be detected, as shown by the white dotted line. The width of the coarse-grained zone was measured to be approximately 2.0 mm.

### 2.2. EP Treatment

After welding, the samples for impact testing were prepared by removing the pass 3, and were then sectioned and ground to small dimensions of 55 × 10 × 5 mm^3^ by electrical discharge machining (EDM). To achieve a complete austenitization, which has been reported to facilitate a considerable enhancement in toughness [11], the Joule heating of EPT is designated to meet three requirements: (1) a maximum temperature above A_C3_, i.e., 847 °C; (2) homogeneous temperature elevation through the thickness, monitored by two points at the center-section, the mid-position A, and outer counterpart B; and (3) a width comparable with HAZ, monitored by sections 2 mm away (monitored by point A1) and 2.2 mm away (monitored by point A2) from the EP-center plane.

For a homogeneous temperature distribution, the finite element method of the ANSYS package with the thermo-electric coupled element Tet 10node 227 was performed to guide EPT configuration. Based on trial results, i.e., a current density *j* = 1400 A/mm^2^, four Joule heating strategies were simulated, as shown in Figure 3: (i) a single pulse with different durations. Figure 3a shows strong concave-shaped isotherms, confirming the electro–contraction effect. Increasing pulse duration can relieve the electro–contraction effect while broadening the spanned zone, which disables EPT for a local problem; (ii) a preheating pulse with lower current density and longer duration; (iii) identical pulse duration with various pulse periods, $t_{p}$. As shown in Figure 3b,c, both strategies can relieve the electro–contraction effect to a certain degree while the requirement (2) can hardly be reached until the dynamic steady temperature distribution was established after certain pulses’ accumulation; and (iv) forced convection with shorter $t_{p}$. Accelerated cooling can be achieved by using blower and copper electrode with a cross-section of 5 × 10 mm^2^. In this regard, three sub-cases with different pulse periods—0.035, 0.04, and 0.05 s—were compared; as shown in Figure 3d, the isothermals at 400 °C become convex as contrasted with the concave shape in the former three cases, suggesting accelerated cooling is an effective routine to prevail over the electro–contraction effect.

The temperature histories at the monitored points are shown in Figure 4. After considerable pulses, the temperature converges dynamically.

Table 1 shows that EP period $t_{p}=$ 0.035 s corresponds to slight temperature inhomogeneous between A1 and B1 as compared to the other two sub-cases $t_{p}=$ 0.05 s and $t_{p}=$ 0.04 s. The maximum temperature at A1 is above A_C3_, meeting requirement (1). The relatively broad width of the heat affected zone is acceptable as a trade-off.

The parameters in EPT were thus determined, as listed in Table 2. Then, experimental EPT was carried out. The typical macromorphology of the EP-treated samples are shown in Figure 3d. The fusion line was revealed as a boundary of weld bead and HAZ, and the arc-shaped morphology presents an isothermal line, validating the numerical results of EPT.

Before EPT, a 2% nital solution was used to etch the samples and reveal the fusion line and HAZ. EPT equipment was a current pulser (MIYACHI is −800 A), as shown in Figure 5. The EP parameters were programmed through the controller and were monitored by an oscilloscope.

### 2.3. Microstructural Characterization and Mechanical Properties

V-shaped notches were machined along the fusion line for the as-welded samples and the EP centerline for the EP-treated samples following the rule of the China Classification Society [12]. Because the samples were small standards, a pad was placed on the impact tester to align to the center of impact load. Charpy impact tests were performed at −60 °C, and the impact toughness values were collected. The fracture morphology was examined by scanning electron microscopy (SEM, 3700) at 5.0 kV.

Metallographic specimens were cut from another side of the HAZ. The specimens were ground with 600, 1200, 2500, and 4000 grit emery paper and then etched with 4% nital solution. The samples for electron backscatter diffraction (EBSD) were mounted, ground, and polished according to the standard metallography procedure. Microscopic observations and measurements of the dislocation density (DD) and geometrically necessary dislocation (GND) were performed using EBSD at an accelerating voltage of 20 kV with a step size of 0.06 μm.

The equivalent grain diameter *D* was calculated based on the EBSD data using the equation $D=2\cdot \sqrt{n\cdot l_{1}\cdot l_{2}/\pi }$, where *n* is the total number of pixels within the grain, and $l_{1}$ and $l_{2}$ are the vertical and horizontal step sizes of the EBSD scans, respectively. The GND density, $\rho_{GND}$, was counted according to the equation $\rho_{GND}=2\theta /ub$ [13], where θ is the local misorientation, *b* is the Burgers vector (0.248 nm for ferrite), and u is the unit length (100 nm) of the point.

## 3. Results

### 3.1. Microstructure Evolutions

Figure 6 shows the inverse pole figures (IPFs) and corresponding boundary distribution. For comparison, Figure 6a,b show the as-welded microstructure and its boundaries distribution, respectively. Two different microstructures, HAZ and weld bead, separated by the fusion line (FL), were observed. It was obvious that the HAZ was coarse-grained. Figure 6c,e show the EP-treated microstructure at A and A1 with full martensitic matrix, suggesting that a through-thickness microstructure evolution from pearlite to martensite was achieved under the designed EPT strategy.

However, the microstructure at A2 was checked to be a spherical pearlitic morphology. The spherical pearlite is usually observed after a thermal cycle below Ac_1_. Similar results can be found in [5], in which the effect of EP on pearlite morphology was investigated. In view of A2 being 2.2 mm away from EP centerline (shown in Figure 3d), the thermal cycle below A2 suggests that the effective width produced by EPT was comparable with HAZ. Considering EP-introduced concentrated energy density and Figure 3d, a conclusion could be made that an effective span above Ac_1_ generated by EPT is within 4 mm.

Figure 7 shows grain size distributions of the as-welded and EP-treated microstructures. The EP-treated distribution shows a more homogeneous distribution.

The strength increment $∆\sigma_{G}$, due to grain refinement and matrix evolution, can be calculated as follows:(1)$$∆\sigma_{G}=∆\sigma_{0}+∆\sigma_{GBs},$$
where $∆\sigma_{0}=\sigma_{m0}-\sigma_{p0}$, $\sigma_{m0}=$ 418 and $\sigma_{p0}=$ 395 MPa are internal friction stress for martensitic matrix and pearlite matrix, respectively. The strength increment due to lattice evolution is therefore calculated to be $∆\sigma_{0}$ = 23 MPa. $∆\sigma_{GBs}$ is the strength contribution from grain boundaries, which can be described using the classic Hall–Petch equation [14,15,16]:(2)$$∆\sigma_{GBs}=\left(k_{p}d_{p}{}^{-1/2}-k_{m}d_{m}{}^{-1/2}\right),$$
where $k_{m}$ = 491 and $k_{p}$ = 464 MPa μm^1/2^ are the strengthening coefficients for the martensitic and pearlitic matrix [16,17], $d_{p}$ and $d_{m}$ are the average grain sizes, measured to be 2.46 μm and 5.05 μm for the EP-treated and the as-welded, respectively. $∆\sigma_{GBs}$ is therefore calculated to be 110 MPa.

### 3.2. Dislocation Hardening Behaviors

The GND evolution of the HAZ extracted by EBSD is shown in Figure 8. Figure 8a shows GND mapping of the as-welded case with an average density of 2.10 × 10^14^ m^−2^, a quantity comparable to that in the martensitic matrix [18], and to the DD of a ferrite matrix with a scale of 10^14^ [19]. Figure 8b exhibits GND mapping of the EP-treated case with an average density of 3.46 × 10^14^ m^−2^, more than 0.6 times higher than that of the as-welded. This GND enhancement is more distinct in terms of accumulated fraction, as shown in Figure 8c.

The contribution of dislocation density to stress increment can be quantified using the classical Taylor relationship [20]:(3)$$∆\sigma_{dis}=\alpha M\mu b\cdot ∆\rho^{0.5},$$
where α and M are an empirical constant related to the dislocation structures and the Taylor factor, 0.435 and 2.75 for steel [20], respectively. μ is the shear modulus with a value of 80.3 GPa and ρ is the dislocation density.

The DD fractions of the as-welded and the EP-treated were measured to be 0.29 and 0.26, respectively, corresponding to a reduction of 10% by EPT. As compared to the increase in GND, the reduction in DD was negligible. Therefore, the ρ in Equation (3) can be estimated according to $\rho =\sqrt{\rho_{D}{}^{2}+\rho_{GND}{}^{2}}$ [21], where $\rho_{D}$ and $\rho_{GND}$ are the dislocation density and GND density, respectively. The $∆\sigma_{dis}$ was therefore calculated to be 40 MPa.

### 3.3. Mechanical Properties

#### 3.3.1. Hardness

The changes in sample hardness before and after EPT were collected. Figure 9 shows a softening of about 12% to 247 HV after welding, comparable to the 10% reduction with a similar hardness level of steel [1]. The softening width was comparable to the measured coarse-grained zone. After EPT, the mean hardness increased to 298 HV, which is 1.2 times that of the as-welded.

The increment in strength $∆\sigma_{y}$ is the sum increment of grain size strengthening ($∆\sigma_{G}$) and dislocation strengthening ($∆\sigma_{dis}$) by ignoring solidification solid strengthening and precipitation strengthening, as follows:(4)$$∆\sigma_{y}=∆\sigma_{G}+∆\sigma_{dis}\text{.}$$

For steel with a strengthen between 300 and 1700 MPa, the relation between hardness and strength can be described as [22]:(5)$$\mathrm{HV}=\frac{1}{2.876}\left(\sigma_{y}+90.7\right)\text{.}$$

According to Equations (4) and (5), the decrease in hardness was calculated to be 30.8 HV due to a coarsened microstructure from 1.41 μm to 5.05 μm by welding, which is comparable with the experimentally measured 33 HV, as listed in Table 3. The increase in hardness due to EPT was estimated to be 58 HV, which was comparable to the experimental measurement of 51 HV. This hardening behavior was opposite to softening behaviors introduced not only by ambient EP [6], where ~40% decrease in hardness has been reported due to electro-plasticity, but also by general tempering treatment due to thermo-plasticity. This result of hardening was unexpected since both electro-plasticity and thermo-plasticity were embedded in this thermal cycle EPT.

#### 3.3.2. Toughness

Figure 10 shows the results of impact tests. A reduction in toughness to 34.7 J was observed after welding. The toughness of the EP-treated samples reads 72.9 J, 2.1 times higher than that of the as-welded.

Typical fracture surface morphologies of the Charpy V-notch impact samples are presented in Figure 11. Figure 11a exhibits two different morphologies: dimple of the weld-bead region and cleavage facet of the as-welded HAZ region. According to the Griffith theory [23], the cleavage fracture stress can be estimated by $\sigma_{c}\propto d^{-1/2}$. Here, *d* can be regarded as the maximum diameter of the coarse grain, which was measured to be 20.35 μm.

Figure 11b shows the dimple fractography after EPT, suggesting the brittle factor was eliminated. Compared to dimples in the weld-bead region, the dimples in the EP-treated HAZ region were coarser, which is consistent with grain size distribution. The coarsest grain was refined to 12.95 μm after EPT. Fine microstructure inhibits cleavage fracture by consuming most of the cracking energy via void coalescence [24]. On the other hand, the fraction of the high angle grain boundary (HAGB) was measured to increase to 34.2% from 15.8% after EPT. A similar increase in HAGB has also been observed for interstitial free steel after EPT [25]. Boundaries of the martensitic packet and block are known to be HAGB and effective in arresting micro-cracks [2]; the increase in HAGB was therefore ascribed to toughness enhancement. Thus, EP-related factors, such as the microstructure refinement and the increase of HAGB, contribute to higher impact-absorbed energies at −60 °C.

## 4. Discussion

### 4.1. Microstructure Evolution Mechanism under EPT

The full martensitic matrix shown in Figure 6c confirms an evolution from ferrite and pearlite shown in Figure 2b via complete austenitization above A_C3_. This change in the total Gibbs free energy can be described as [26]:(6)$$∆G_{EPT}=∆G_{0}+∆G_{e},$$
where $∆G_{0}$ is the change of free energy in a conventional thermal system, $∆G_{e}=\mu_{0}g\xi j^{2}\Delta V$ is the energy change generated by electric distribution, where $\mu_{0}$ and ΔV are the magnetic susceptibility in a vacuum and the volume of a nucleus, respectively, *g* is a positive geometric factor, ξ is a factor, ξ<0 for austenite with less crystal deformation [27], then $∆G_{e}<0$, which yields:(7)$$∆G_{EPT}<∆G_{0}\text{.}$$

Equation (7) indicates that EP decreases the thermodynamic barrier of austenitization, and thus facilitates the nucleation rate as compared to conventional thermal treatment [7], which is therefore a contributor to the microstructure refinement shown in Figure 7.

There is a very short time allowing for atomic diffusion. According to the Arrhenius equation: D = D_0_exp(−Q/RT), where D is the actual diffusion rate, D_0_ is the initial diffusion rate and Q is the activation energy of diffusion. R is the gas constant. At the average temperature of 828 ℃ shown in Figure 3a, carbon diffusion rate in ferrite was calculated to be 9.9 × 10^−11^ m^2^ s^−1^ with D_0_ = 6.2 × 10^−7^ m^2^ s^−1^ and Q = 80 kJ mol^−1^ [28]. Since the mean HAZ grain size is 5.05 μm, the time need for the carbon atom to cross over was calculated to be 0.043 s according to $X_{3D}={\left(6Dt\right)}^{-1/2}$ [29], far less than the EPT time of 0.59 s, which validated the occurrence of a full austenizating during EPT.

### 4.2. Hardening Mechanism through GND

The unexpected hardening due to GND shown in Figure 8 makes understanding the mechanism necessary. However, the result is multi-factor-coupled, including the phase transformation from pearlite to martensite, which hampers a deep understanding of EPT hardening. To decouple this effect, the martensite-based matrix, i.e., weld bead, will be focused on in this subsection. Three sub-cases listed in Table 1 were carried out. In fact, EPT introduced a localized electro- and thermo-cycle, which generated a competition between plasticity and work-hardening. Those plasticity components were correlated with the hardening mechanism in terms of plasticity behaviors, i.e., dislocation, dislocation-line and gradient dislocation (i.e., GND).

#### 4.2.1. Dislocation

Since the plastic strain was experimentally inaccessible, numerical simulation was utilized here as an alternative through thermo-mechanics in the ANSYS package by coupling the thermal results listed in Table 1. The thermo-mechanical constitutive model of FH690 steel was referred to in [30]. The simulated plastic strain at point A in three EP sub-cases was extracted and is shown in Table 4.

Firstly, work-hardening was checked by correlating with EP-introduced DD evolution. DD evolution introduced by EPT in three-subcases show a tendency of increasing first and then decreasing with increases in plastic strain, as observed in Figure 12 in the solid line. While DD introduced by work-hardening alone shows a monotonous ascendant tendency, as calculated according to the Ashby model, $\rho_{\varepsilon D}=\frac{4f}{b}\cdot \frac{\varepsilon }{r}$ [31], where *f* is the fraction of martensite, *r* is the radius of a spherical martensite. This inconsistency between the EPT-introduced and the Ashby model indicates that DD evolution is not work-hardening dominant alone.

Then, electro-plasticity introduced by drifting-electron was linked with DD evolution. With a current density far beyond reported threshold value 7.4 A/mm^2^ for high strength steel [8], the drag force $F_{j}$ by drifting-electron can be calculated according to [32]:(8)$$F_{j}=\frac{{\left(j\Delta A\right)}^{2}}{n_{c}v_{e}}R_{C}\approx \frac{M_{Fe}\gamma_{c}n_{e}e}{\rho_{d}}j=K_{e}j,$$
where $M_{Fe}$ is the molar mass of iron, $n_{e}$ is the charge carrier density, which is the conduction electron density in metals, and for iron, $n_{e}$ = 1.7 × 10^29^ m^−3^, $\gamma_{c}$ is the specific resistivity of carbon atoms having the value of 5.9 × 10^−8^ Ωm/at, $\gamma_{c}$ and $v_{e}$ are mass density and electron drift velocity, respectively, and $K_{e}$ is the electron wind force coefficient with a value of 0.011 MPa/(A·mm^−2^) [33]. $F_{j}$ was calculated to be 15.4 MPa, lower than thermal stress $\sigma_{A}$, 62.1 MPa (shown in Table 3). Moreover, *F_j_* becomes lower in temperature [34], indicating that electro-plasticity is descendent-dependent with temperature, as schematically shown in Figure 13 by the dotted line.

At last, as regards the thermo-plasticity, it is generally accepted that DD is a monotonous descendent, dependent on increasing temperature when tempering, as schematically shown in Figure 13 by the centerline.

Note that the increase in *t_p_* corresponds to the decrease in temperature and plastic strain. Combined with the dependent tendencies and relatively quantitative relations discussed above, the competition relation among three plasticity components, electro-plasticity D(j), thermo-plasticity D(T) and work-hardening D(ε), as schematically depicted in Figure 13 can be concluded. Regions I and III correspond to softening regions, dominated by D(T) and D(j), respectively. Region II is the working-hardening dominant which compensates or prevails over the softening due to the thermo- and electro-plasticity, which is thus favorable for hardening. It seems that EPT parameters should be properly scheduled since the window for favorable hardening is narrow. However, this prediction of this narrow window calls for more quantitative work about the dependent models of the three plasticity components which are scarcely observed in the literature.

#### 4.2.2. Dislocation Line

The dislocation line is known to be grain boundaries with misorientations ranging from 2° to 15°, and also be a martensitic lath substructure. The critical stress to initiate the dislocation line can be expressed as τ=μbL, where *L* is identical to the lath width. For *L* < 0.1 μm, the critical stress was estimated to be as high as 199 MPa, which is therefore known as immobile dislocation (IMD) under conventional thermal conditions [35]. However, this phenomenon was not followed under EP, as shown in Figure 14. The fraction of IMD was measured to decrease by 19% with respect to either plastic strain or grain size. A similar tendency of IMD against plastic strain has also been observed in ambient EP [6,8]. This monotonous decrease of the IMD was therefore attributed to the dominant electro-plasticity by excluding work-hardening with an enhancing effect.

#### 4.2.3. Gradient Dislocation

The GND evolutions between the cases of *t_p_* = 0.05 s and 0.035 s were compared, as shown in Figure 15. The GND evolutions were actually undetectable, as observed from Figure 15c.

Note that it is generally accepted that the GND is grain size-dependent, the GND evolution was then linked with grain size. Figure 16 shows a descendent tendency similar to the annealing [36] and the tempering [37]. Supposing the GND follows the grain-size dependent tempering [37], the grain-size refinement in Figure 8 would produce a GND increase of 42%. However, this estimated increase is still less than the measured 60%, as shown in Figure 8. Thus, the work-hardening was also a contributor to the measured GND enhancement, suggesting that the GND enhancement is work-hardening dominant by prevailing over the thermo- and electro-effects.

In summary, this thermal cycle EP imposes an influence on the dislocation structures in a selective mode, slightly degrading in DD and IMD due to the prevailing thermo- and electro-effects, and largely enhancing GND due to dominant work-hardening and microstructural refinement. This selectivity tailors martensitic in a way favorable for hardening and toughening: (i) the martensitic laths coalesced into packets and blocks, confirmed by decreased IMD [19]; (ii) the prior austenite boundary broken into packets, as confirmed by refined microstructure; and (iii) dislocation scattered into grain interior [6], confirmed by annihilated DD and elevated GND density. In addition to the increase in HAGB, those tailors contribute to the most improved toughness and less improved hardness.

## 5. Conclusions

EPT was utilized to toughen and harden the HAZ of high strength steel. The local behavior of EPT was detailed in terms of dislocation components. The main conclusions were as follows:(1)Toughening and hardening a thick HAZ with a limited width was a challenge for a homogeneous temperature elevation through thickness due to electro–contraction; EP parameters characterized with surficial forced cooling and multiple pulses with shorter period were a proper solution;(2)The pearlite matrix HAZ evolved to refine martensitic matrix with less dislocation density, a highly geometrically necessary dislocation density and a high fraction of high angle grain boundary. This selective tailoring by this local thermal cycle EP gave rise to a 2.1 times and 1.2 times increase in toughness and hardness, respectively;(3)This local thermal cycle EP introduced an interaction among three plasticity behaviors, thermo- and electro-plasticity and work-hardening. The former two are dominant in the evolution of DD and IMD, while the third is dominant in GND evolution;(4)The microstructural tailor spanned less than 4 mm for 5 mm thick samples.

## Figures and Tables

**Figure 1 materials-15-05847-f001:**
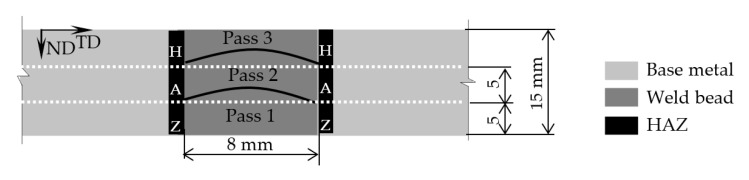
Schematic of HAZ configuration.

**Figure 2 materials-15-05847-f002:**
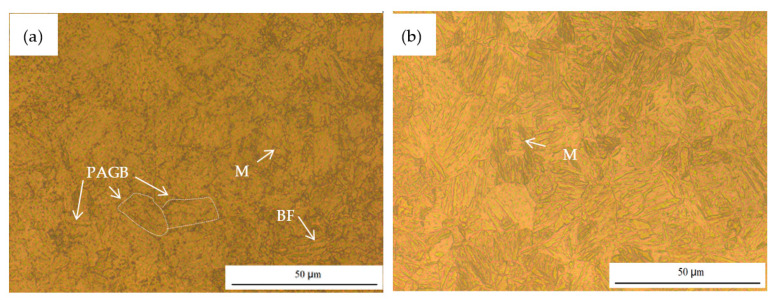
Optical micrographs of (**a**) heat-affected zone and (**b**) weld bead.

**Figure 3 materials-15-05847-f003:**
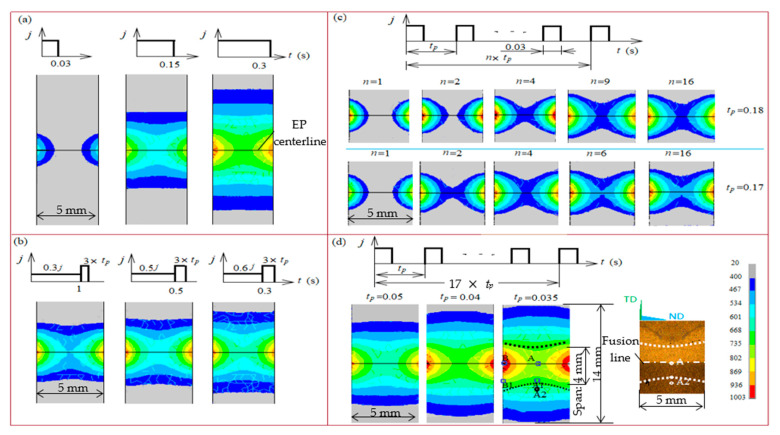
Temperature contour maps for (**a**) single EP; (**b**) preheat EP; (**c**) various *t_p_*s of EP and (**d**) various *t_p_*s of EP subjected to accelerated cooling.

**Figure 4 materials-15-05847-f004:**
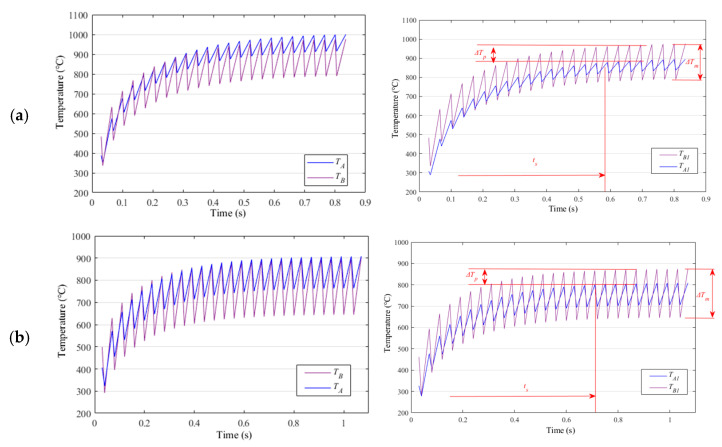
Temperature history at sampled points at various pulse periods, $t_{p}$ (**a**) 0.035; (**b**) 0.040 and (**c**) 0.050 s.

**Figure 5 materials-15-05847-f005:**
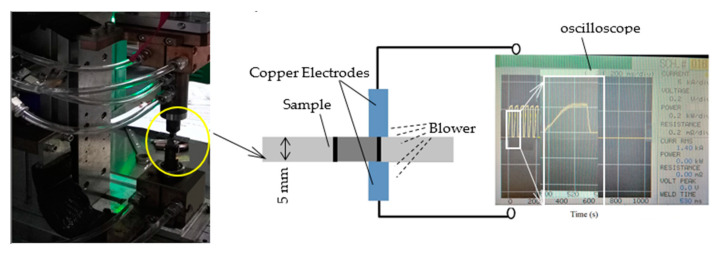
EPT test setup.

**Figure 6 materials-15-05847-f006:**
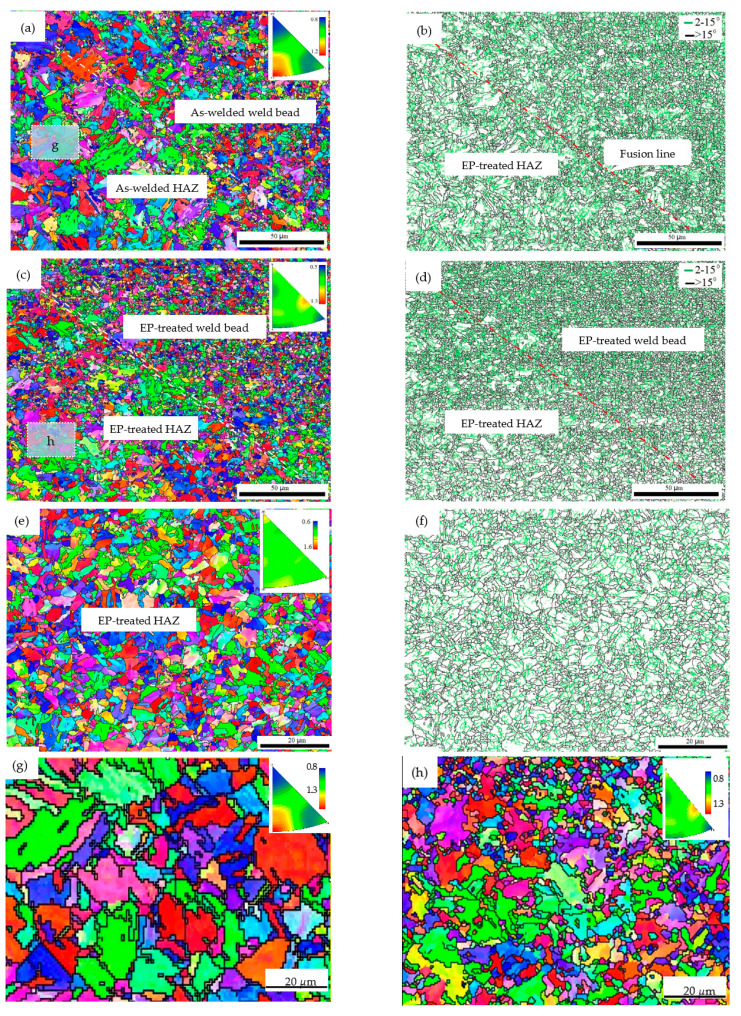
EBSD inverse pole figures (IPF) and boundary distributions of the (**a**,**b**) as-welded, (**c**,**d**) EP-treated A and (**e**,**f**) A1, respectively, and IPF figures (**g**,**h**).

**Figure 7 materials-15-05847-f007:**
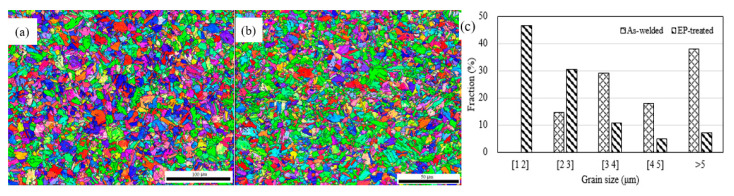
The (**a**) as-welded and (**b**) EP-treated microstructures, and (**c**) grain size distributions of (**a**,**b**).

**Figure 8 materials-15-05847-f008:**
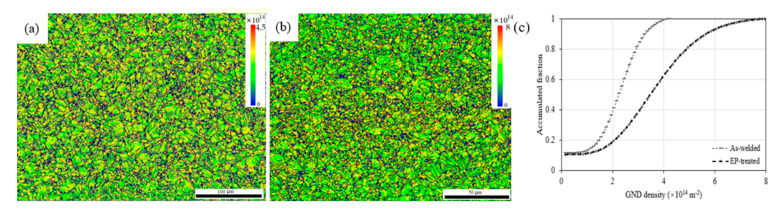
GND mapping of the (**a**) as-welded, (**b**) EP-treated and (**c**) accumulated fraction.

**Figure 9 materials-15-05847-f009:**
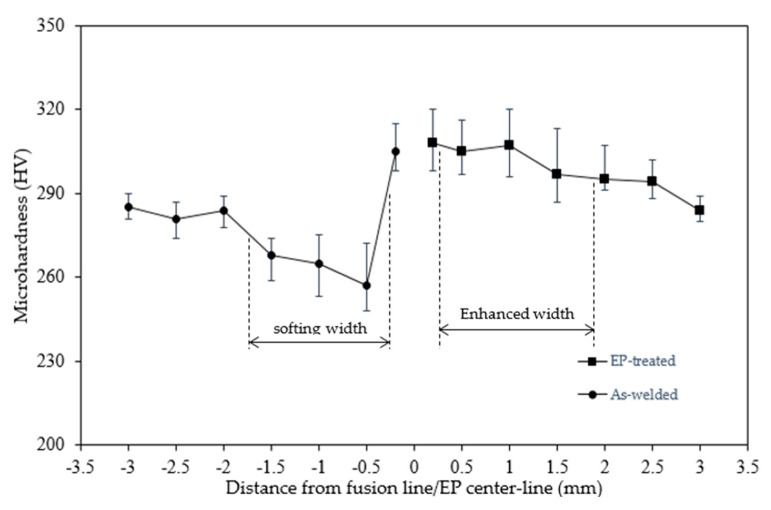
Microhardness of the EP-treated and the as-welded microstructures.

**Figure 10 materials-15-05847-f010:**
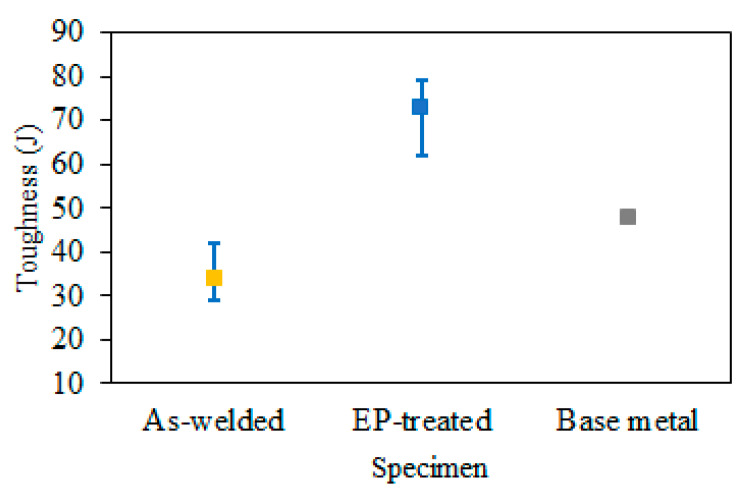
Evolution of Impact toughness.

**Figure 11 materials-15-05847-f011:**
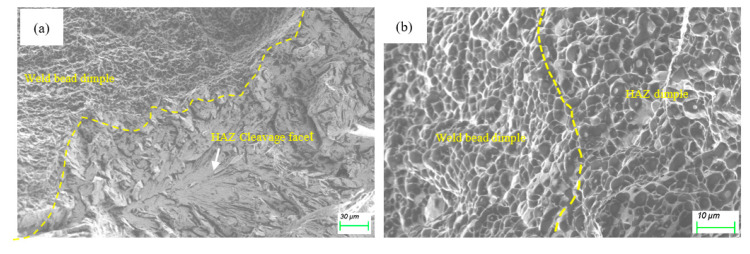
Fractograph of (**a**) the as-welded and (**b**) the EP-treated.

**Figure 12 materials-15-05847-f012:**
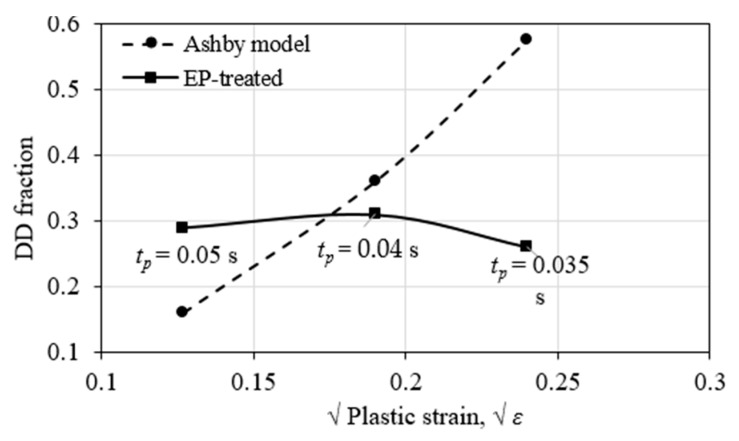
Comparison of DD evolutions between EPT-hardening and work-hardening.

**Figure 13 materials-15-05847-f013:**
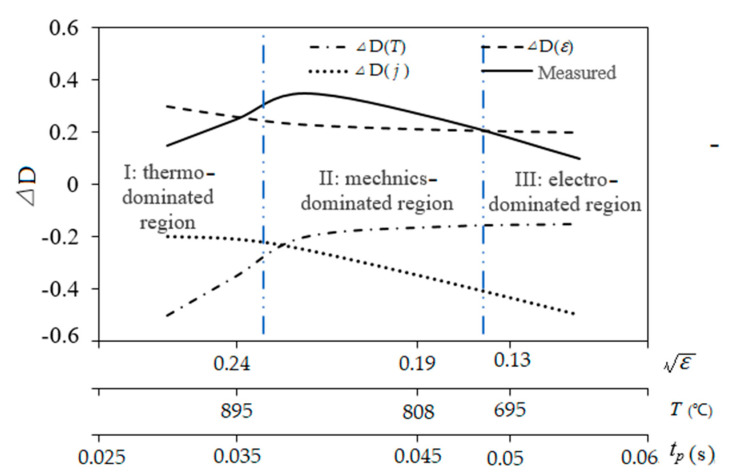
Schematic competition among plasticity components under EPT.

**Figure 14 materials-15-05847-f014:**
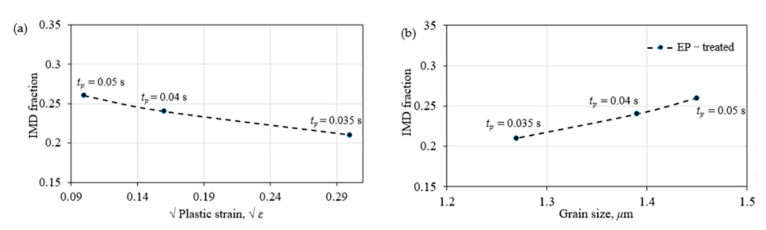
IMD evolutions against (**a**) strain and (**b**) grain size.

**Figure 15 materials-15-05847-f015:**
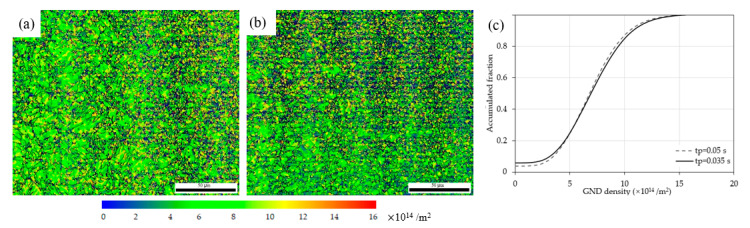
GND mapping of (**a**) *t_p_* = 0.05 s, (**b**) *t_p_* = 0.035 s and (**c**) accumulated fraction of (**a**,**b**).

**Figure 16 materials-15-05847-f016:**
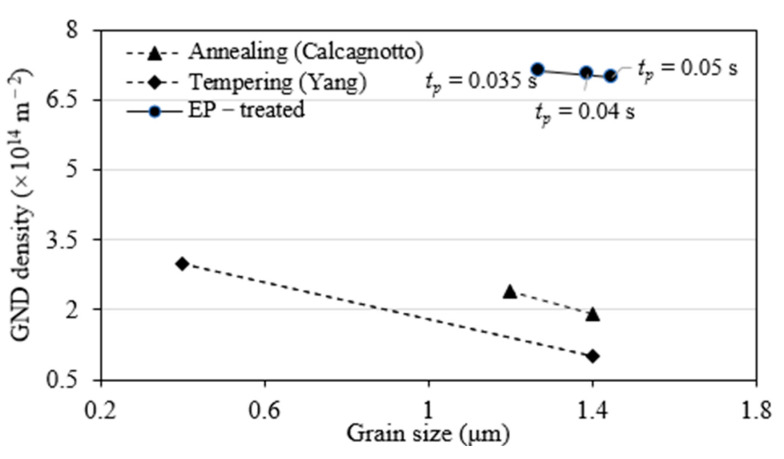
GND evolution under EPT.

**Table 1 materials-15-05847-t001:** Detailed results of Joule heating.

Subcases $t_{p}$ (s)	Temperature Difference $∆T_{p}$ between A1 and B1 (℃)	Max. Temperature at A (℃)	Max. Temperature at B (℃)	Half-Width Experiencing Thermal Cycle above 400 ℃ (mm)	Total EPT $t_{s}$ (s)
0.035	55	895	1000	7.0	0.59
0.040	66	808	907	5.7	0.66
0.050	92	695	804	4.6	0.84

**Table 2 materials-15-05847-t002:** Parameters in EPT.

Density. *j* (A/mm^2^)	Pulse Period. $t_{p}\;(s)$	Pulse Duration. $t_{d}\;(s)$	Pulse Number. *n*	EPT Time $t_{s}\;(s)$
1400	0.035	0.03	17	0.59

**Table 3 materials-15-05847-t003:** Hardness evolutions due to welding and EPT.

Process	ΔHVex (Experimental)	ΔHVes (Estimated)	Matrix Evolution	Matrix Refinement	DD	GND
As-welded	−33	−30.8	-	−30.8	-	-
EP-treated	51	58	7	37	−0.1	14

**Table 4 materials-15-05847-t004:** Thermo-mechanical result at point A.

*t_p_* (s)	Temperature °C	*ε_A_*	*σ_A_* (MPa)
0.035	895	0.0593	62.1
0.04	808	0.0361	75.9
0.05	695	0.016	117.0

## Data Availability

Data sharing not applicable.

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
