# Peer review of "Toughening and Hardening Limited Zone of High-Strength Steel through Geometrically Necessary Dislocation When Exposed to Electropulsing"

_materials, 2022, doi:10.3390/ma15175847_

Round 1
Reviewer 1 Report
The authors present an interesting paper titled: “Toughening and hardening limited zone of high-strength steel through geometrically necessary dislocation when exposed to electropulsing”
The paper develops different points; nevertheless, some of them must be modified and retaken in order to be publishable this manuscript:
1) The manuscript must be rearranged and synthetized in order to present with clarity the main content of the paper.
2) Figure 1: The characters are small; they must be larger. The thickness of the sample seems of 15 mm before the EP treatment, and of 5 mm in thickness for the EP treatment; there is not a temperature gradient effect during these processes ?
3) For the experimental results, it should be stated clearly : the applied number of pulses, the period and the relationship with : the nucleation, dislocations, and accumulation of dislocations on the sub-grain boundaries.
4) What precipitates were generated during the ET process ?
5) Figure 6. It is not indicated information concerning : pulses, period,…
6) Equations must be listed, separated from the manuscript.
7) It is recommended to generate a Table with the main parameters of welding and EP process, such as: current, voltage, temperature, current density, pulse period ant its influence on the two measured mechanical properties : toughness and hardness.
8) Conclusions section must be expanded, according with the content of manuscript.
9)In some sections, English must be revised.
Reviewer 2 Report
This paper explores the possibility of using electropulsing to alleviate the deleterious microstructure caused by fusion welding. Microstructure and mechanical properties are then given. The conditions of electropulsing are unclear, it looks like maybe two electrodes are on either side of the sample. Electrode materials and dimensions are not given in this paper.
Many portions of this paper are directly copied from this reference: 10.3390/ma15062135 Including sentences word-by-word changed just ever so slightly occasionally to bypass automatic plagerism checks. This is inappropriate. This paper is likely to be a continuation of the reference material, but it is not written as such.
The figures are poor quality. The axes are too small to be easily read. Units are missing on figure 2. Length scales are unreadable in Figure 4. Contrast is very low in Figure 4. They describe martensite, but it is not obvious. Some x-ray diffraction would help in determining whether it remained FCC or not, but this optical micrograph is inadequate. The labels are inconsistent in color.
Figure 5 is completely messed up. There are some major concerns with the EBSD mapping and grain size analysis. Many of the smaller grain sizes appear to be less than 5 pixels across. There seems to be too much cleaning up or too large of a spot size.
If I understand Figure 5 correctly, the microstructure should be symmetric across the fusion line, but one side has larger grains than the other. The EDS in figure 9b is completely pointless combined together like that.
Figure 13 a "thermos-treatment" is never described.
This paper has an interesting relationship between "dislocation density" and "geometrically necessary dislocations." and how GND's increase while DD. But aren't GND's DD's? Perhaps the strain you are measuring in EBSD data are not what the paper suggests it is providing. This may also explain the strong departure from the Ashby model in Figure 10. Rereading this, I take serious issue with the DD and GND. It appears that DD is calculated from mechanical behavior and GND is calculated from EBSD. The two techniques cannot be mixed and matched to provided different results. Extra characterization including x-ray diffraction or transmission electron microscopy is required to make the claims made in this paper.
Conclusion point 4 makes no sense. It should be rewritten without the word "polluted."
Due to the constant poor figure quality and inadequate characterization, I recommend the paper to be rejected. The topic is interesting and I encourage the authors to continue this work until it is at a higher quality.
Reviewer 3 Report
General comments
The aim of this study is to investigate the enhancement in both low-temperature impact toughness and hardness of high strength steel heat-affected zone by using high-density electropulsing.
The topic is interesting and well written and fits within the scope of the journal.
The Referee recommends the publication on the journal after a minor revision of the paper.
Specific comments
Some comments relative to each section are below reported.
1) The reviewer suggests to introduce the unit measure of temperature in Figure 9.
2) The reviewer suggests to add further information about the model developed in Ansys. What element are used? What type of mesh is used? Etc
3) The reviewer suggests to introduce the number and reference of the China Classification Society
4) For the reviewer opinion is not clear if the authors have done Charpy tests or they have taken the image described in section 3.3 from literature. Please clarify it.
Round 2
Reviewer 1 Report
The authors have made the main suggested modifications and improvements.
Author Response
We appreciate for reviewer’s comments!
Reviewer 2 Report
Most of the direct copies are now modified, but the very first sentence of both papers is still word for word.
From the most recent draft: "High-strength steel (HSS) is a promising material for lightweight applications, especially in oceanic engineering, where mechanical properties such as high strength and good toughness are required."
From https://doi.org/10.3390/ma15062135, "High-strength steel (HSS) is a promising material for lightweight applications, especially in oceanic engineering, in which the mechanical properties of high strength and good toughness are required."
Figure 2 is still poor. Martensite can be discerned, but not pearlite or ferrite. the PAGB is not obvious as the paper claims.
I will allow for the explanation of dislocation content now that it is clearer. The collection of EBSD data is still poor and unaddressed. The paper is collected at about the same resolution per pixel as the sizes of GB are being reported. Grain size can hardly be determined without large error and GND measurements are a cause for even more error. While a large scan may be necessary to understand position, A scan with better resolution is also required. If the authors feel the low-magnification EBSD is necessary, then both should be included. IPF orientation keys should be included also.
The scale bars on figure 8 are not consistent and can't be compared against each other easily.
So much of this paper hinges on the EBSD which is not taken in good enough resolution, I believe I am still forced to reject it.
